# Characteristics, Cryoprotection Evaluation and In Vitro Release of BSA-Loaded Chitosan Nanoparticles

**DOI:** 10.3390/md18060315

**Published:** 2020-06-15

**Authors:** Qinying Yan, Jiaqi Weng, Xieqi Wu, Weiwei Wang, Qingliang Yang, Fangyuan Guo, Danjun Wu, Ying Song, Fan Chen, Gensheng Yang

**Affiliations:** 1College of Pharmaceutical Sciences, Zhejiang University of Technology, 18 Chaowang Road, Hangzhou 310032, China; yqy@zjut.edu.cn (Q.Y.); wengjiaqi0088@126.com (J.W.); wuxieqi1994@163.com (X.W.); wangweiwei199504@163.com (W.W.); qyang@zjut.edu.cn (Q.Y.); guofy@zjut.edu.cn (F.G.); danjunwu@zjut.edu.cn (D.W.); songying@zjut.edu.cn (Y.S.); 2Hubei Collaborative Innovation Center for Green Transformation of Bio-Resources, Life Sciences School of Hubei University, 368 Youyi Road, Wuhan 430062, China; chenfan1688cn@aliyun.com

**Keywords:** characteristics, cryoprotection evaluation, dynamic high pressure microfluidization, chitosan nanoparticles

## Abstract

Chitosan nanoparticles (CS-NPs) are under increasing investigation for the delivery of therapeutic proteins, such as vaccines, interferons, and biologics. A large number of studies have been taken on the characteristics of CS-NPs, and very few of these studies have focused on the microstructure of protein-loaded NPs. In this study, we prepared the CS-NPs by an ionic gelation method, and bovine serum albumin (BSA) was used as a model protein. Dynamic high pressure microfluidization (DHPM) was utilized to post-treat the nanoparticles so as to improve the uniformity, repeatability and controllability. The BSA-loaded NPs were then characterized for particle size, *Zeta* potential, morphology, encapsulation efficiency (EE), loading capacity (LC), and subsequent release kinetics. To improve the long-term stability of NPs, trehalose, glucose, sucrose, and mannitol were selected respectively to investigate the performance as a cryoprotectant. Furthermore, trehalose was used to obtain re-dispersible lyophilized NPs that can significantly reduce the dosage of cryoprotectants. Multiple spectroscopic techniques were used to characterize BSA-loaded NPs, in order to explain the release process of the NPs in vitro. The experimental results indicated that CS and Tripolyphosphate pentasodium (TPP) spontaneously formed the basic skeleton of the NPs through electrostatic interactions. BSA was incorporated in the basic skeleton, adsorbed on the surface of the NPs (some of which were inlaid on the NPs), without any change in structure and function. The release profiles of the NPs showed high consistency with the multispectral results.

## 1. Introduction

Therapeutic protein has attracted extensive attention in recent years due to its tremendous therapeutic benefits in treating widespreading and severe health problems such as cancer, infectious diseases, autoimmune diseases, AIDS/HIV, and related conditions [1,2]. Unfortunately, most proteins are highly vulnerable, as their activity can be strongly influenced in vitro by many factors such as pH, ionic strength, temperature, high pressure, non-aqueous solvents, etc. Additionally, proteins also can be easily degraded by enzymes and proteases in vivo [3], not only at the administration site but also en route to the site of pharmacological action, resulting in a reduced bioavailability.

For these reasons, competent drug delivery systems, typically nanoparticles (NPs), are commonly utilized to load these proteins as cargos, potentially producing essential benefits, such as enhancing solubility and thermodynamic stability of proteins, controlling the release of protein molecules, improving biodistribution and prolonged half-lives of proteins, as well as targeting the diseased tissue in vivo [4,5]. In fact, different drug-loaded NPs have their unique advantages. For example, anticancer protein-loaded NPs are able to penetrate tumors due to their small particle size, and the leaky nature of tumor microvasculature [6,7]. For infectious diseases, NPs are often used as vehicles for co-delivery of antigen and immunostimulatory molecules to the same APC, which can promote cross-presentation and help elicit a broader cellular response [8]. In order to reduce the cytotoxicity of inorganic nanoparticles during drug administration and to expand the potential applications of nanomedicine technology, plenty of researches have been extensively invested into. More recently, various natural biomaterials, especially polysaccharides, have been greatly introduced and developed for protein-loaded NPs, because of their outstanding physical and biological properties. Chitosan (CS), the only natural cationic biopolymer, is one of the most up-and-coming biopolymers among all the natural polymers and [9] is widely applied because of remarkable properties such as biocompatibility, non-toxicity, biodegradability, and low immunogenicity. Most CS are present in the cuticles of arthropods, especially in the shells of marine animals such as crabs and shrimps [10]. The free amino groups of CS can spontaneously form NPs with anionic molecules, and this ionic gelation process is relatively mild, avoiding the use of harsh organic chemicals and high temperatures, so that proteins can be successfully encapsulated. It has been reported that Chitosan nanoparticles (CS-NPs) loading proteins (e.g., bovine serum albumin (BSA), ovalbumin (OVA), insulin, antigens, lysozyme) [9] were prepared via the ionic gelation process to obtain high encapsulation efficiency (EE), with no change in the biological activity of proteins, which can be a very promising candidate for drug delivery applications [11].

A large number of studies have been taken on NPs characteristics (e.g., particle size distribution, surface charge) of different therapeutic proteins; very few of these studies have focused on the microstructure of protein-loaded NPs, which may closely relate to the in vivo pharmacokinetics/pharmacodynamics processes [12,13]. For example, high molecular weight chitosan (HMWC) is generally associated with higher EE, because high molecular weight polymer chains facilitate the immobilization of proteins within the chitosan matrix [14,15], accompanied by more complex NP microstructures. Meanwhile, the release was much faster when the loading capacity (LC) was higher, due to the release of proteins mainly located on the NP surface [16]. It is generally recognized that the release mechanisms of proteins loaded in chitosan NP are desorption, dissociation [17], diffusion, and erosion [18,19]. In addition, the mechanism of protein release is greatly affected by the interaction forces and microstructure in NPs, which ultimately affect the pharmacokinetic process in vivo. The conventional methods of nano preparations have numerous disadvantages, for instance, longtime consumption, low yield, low repeatability, and easy contamination. For this reason, nanoparticles have been studied in the laboratory thoroughly but rarely applied to clinical treatment.

In this study, the dynamic high pressure microfluidization (DHPM) was used to address most of the ongoingcritical challenges in amplification reactions during the manufacturing processes of nanoparticle. As shown in Figure 1, the microjet is a fluid witchthrough a valve with a very small aperture under the ultra-high pressure (310 MPa) to achieve dispersion, homogenization, emulsification etc. It is able to expand the fluid handling from merely micro scale down to a more precise nanoscale [20], which is a typical technical equipment developed based on the micro-jet technology to achieve the specific purpose of material emulsification and homogenization. The implementation of nanoparticle production through DHPM was reproducible and highly controllable. In addition, DHPM requires less media volumes, shorter time, and is more efficient compared with traditional amplification reactions [21].

The present study aims to design and develop novel therapeutic protein-loaded NPs with CS, a typical natural biomaterial, to improve the therapeutic effect and bioavailability of such protein. Freeze-drying studies are performed to produce re-dispersible dry powders to improve the stability of BSA-loaded NPs and using the variety of spectroscopy techniques to figure out the interaction forces and microstructure in CS-NPs.

## 2. Results and Discussion

### 2.1. NPs Preparation and Characterization

In our preliminary experiment, low molecular weight chitosan (LMWC), medium molecular weight chitosan (MMWC), and high molecular weight chitosan (HMWC) were chosen to prepare NPs. However, as the molecular weight increased, the viscosity of CS also become higher, leading to some problems in the preparation process. For example, CS cannot pass through the filter membrane to remove the impurities, so we chose LMWC as the material for the preparation of NPs.

LMWC is a natural polymer having a number-average molecular weight (MW) of 166,000 (ranging from 104,000 to 225,000). In previous studies, BSA was encapsulated into NPs through a facile self-assembly approach in which electrostatic interaction of positively charged CS with oppositely charged Tripolyphosphate pentasod (TPP) is the primary driving force for nanostructure formation by a self-assembly process. Preliminary experiments were addressed to discovery the main influencing factor in NP formation. The particle size and *Zeta* potential of the NPs were determined at 25 °C by the dynamic light scattering (DLS) (Zetasizer Nano ZS90, Malvern, UK). Particle size values were reported as the mean hydrodynamic diameter (MHD), standard deviation (SD), and polydispersity index (PDI). Each batch was analyzed in triplicate. During this process, the opalescence was used as an indicator, which was also confirmed by means of dynamic light scattering [22]. As shown in Figure 2A, BSA-loaded NPs suspension was transparent with weak opalescence, indicating the small particle sizes.

Compared with blank NPs (CS: TPP = 3:1), a significant decrease in particle size and PDI were observed in BSA-loaded NPs, showing that BSA was favorable for the formation and stability of NPs, which was consistent with the fluorescence quenching measurement. The effect of CS: TPP weight ratio on particle characteristics is also reported in Table 1. The obtained results illustrated that the increase in weight ratio was accompanied by an increase in particle size and *Zeta* potential. Similar observations have also previously been reported [23]. All NPs showed positive *Zeta* potentials due to protonated amine groups of CS that were responsible for such a positive surface charge. The PDI values of all samples were smaller than 0.3, which suggested the narrow distribution of particle size. EE and loading capacity of “pharmaceutical agents” in NPs are important factors to be considered to evaluate the therapeutic efficacy of nanocarriers by in vitro and in vivo models [24]. As shown in Table 1, the highest EE (52.7%) and LC (66.9%) of BSA were observed in the NPs which CS: TPP (*w/w*) was 3:1. The reduced usage of TPP would lead to a retarded interaction between the BSA and TPP and a reduced encapsulation of BSA into nanoparticles due to the inadequate crosslinker (TPP), consequently, the weight of the formed NPs was too low (lower than 2 mg) to be weighted. Meanwhile, the increase in BSA EE was also dependent on the concentration of the CS in the NPs (Appendix A), suggesting a high affinity of the positively charged CS binding to BSA.

The situation in which the drug LC is greater than the encapsulation rate probably was caused by the calculation of drug LC, which is greatly related to the weight of nanoparticles, yet the value of the electronic balance was inaccurate when the weight of the analyte was less than 2.0 mg. That is why we use microfluidics for amplification experiments to further make the nanoparticles more uniform. The DHPM can massively produce nanoparticles with a PDI values less than 0.2 (20 times of traditional preparation method), and the calculated drug LC through DHPM is more reliable. This is capable to provide theoretical and practical basis for the industrialization and mass production of nanoparticles.

### 2.2. Preparation of the NPs through DHPM

To improve the drug LC with an acceptable uniformity of nanoparticles as well as satisfactory repeatability of experimental results, DHPM was selected for post-treating of NPs. After determining the optimal CS: TPP ratio (*w/w*), according to this ratio, BSA solution (1 mg/mL) in deionized water were added directly to the CS solution, then placed in a high pressure homogenizer for 5 min and TPP solution (0.7 mg/mL) was added to the CS solution dropwise while homogenizing. The nanoparticle crude solution was then placed into the DHPM and circulated under different pressures and different cycle times.

In our study, LMWC (1 mg/mL) was mixed with BSA (1 mg/mL) and dissolved in Tris-HCl buffer (pH 5.5), the CS: TPP (*w/w*) = 3:1 was chosen as the optimal ratio for the preparation of NPs. According to this ratio, BSA solution (1 mg/mL) in deionized water were added directly to the CS solution, placed in a high pressure homogenizer for 5 min and TPP solution (0.7 mg/mL) was added to the CS solution dropwise while homogenizing. The effect of pressures and cycle times on particle characteristics is also reported in Table 2 and Table 3.

As shown in Table 2, compared to the coarse NPs, the EE of those post-treated NPs with DHPM has increased significantly, probably because in the high-pressure environment, the free drug on the surface of the nanoparticle is further adsorbed into the nanoparticles. The drug loading the results of the Table 2 indicated that different pressures had no effect on the particle size, *Zeta* potential and EE of the nanoparticles. As the pressure increased, the drug LC of nanoparticles had a significant increase. Because of the limitations of the instrument, we did not choose a higher pressure. The instrument would be rapidly heated to 80 °C in a short time if the pressure continued to increase, leading to an inactivation of the encapsulated protein drug.

As shown in Table 3, the cycle time had no influence on the particle size, *Zeta* potential, morphology, EE, and drug LC. On the contrary, when increasing the cycle time from 0–8 min, the drug LC firstly increased from 33.4% to 39.8%, indicating that the more cycles, the more drug LC of the nanoparticles. However, further increasing the cycle time to 10 min led to partial desorption of the nanoparticles, which will reduce the drug LC.

The morphology of freshly prepared NPs was observed under transmission electron microscope (TEM), as shown in Figure 2B, which exhibited spherical shape and uniform-size distribution consistent with the DLS measurement (Figure 2C). DLS showed that the particle size of NPs is 137 nm while TEM indicated about 130 nm. This is probably because DLS reflects the hydrodynamic diameter of NPs swelling in aqueous solution, whereas TEM reflects the diameter of dried NPs. Therefore, the diameter determined by TEM was smaller than that measured by DLS [25].

### 2.3. NPs Lyophilization

NPs are often produced in an aqueous suspension form. However, NP suspension is unstable. For example, particle aggregation and fusion, polymers degradation and drug leakage, and growth of microorganisms [26,27] often occur during long periods of storage. Freeze-drying of aqueous suspensions into solid powders has been used to improve the long-term stability of NPs [28]. However, this dehydration process may generate freezing and desiccation stresses on the NPs, accompanied by irreversible aggregation of the NPs. These stresses can be minimized by using cryoprotectants and lyoprotectants, possibly due to the formation of amorphous cryoconcentrated suspension or ice during the freezing or drying steps. The cryoconcentrated phase is constituted by nanoparticles, surfactants, unloaded drugs, etc. A high concentration of nanoparticles may cause aggregation or fusion in the process of cryoconcentrating, and the crystallization of ice may cause mechanical stress to nanoparticles. The excipients were added to protect nanoparticles from freezing stresses [29]. In our study, it was found that most of the NPs without any cryo- or lyoprotectants were aggregated. Therefore, cryo- and lyoprotectants were necessary for the lyophilization process. On the premise of adding cryo- and lyoprotectants, the DLS was used to characterize the nanoparticles before and after lyophilization by the method described in 2.1. The increase of particle size of re-dissolved BSA-loaded NPs after lyophilizing was calculated to evaluate the protective ability of cryo- and lyoprotectants during the lyophilization process of nanoparticles. With the purpose of cryoprotection and cryopreservation for the lyophilized samples, different sugars such as trehalose, glucose, sucrose, and mannitol were used [30]. NPs lyophilized with mannitol could not completely re-dissolve in water [12]. Re-dispersibility is defined as the ability of dried NPs to recover to their original state in an aqueous system, evaluated in terms of particle size, PDI, and morphology. After lyophilization, NPs with sucrose, glucose or trehalose as cryoprotectant exhibited almost common behavior when the concentration was 10%, and the particle size of those were all smaller than 200 nm (Figure 3A). With the increase of mass ratio of cryoprotectant to NPs (5:1 to 10:1), the particle size of lyophilized samples was smaller, which was similar to newly synthesized NPs. Re-dissolved BSA-loaded NPs were transparent in their appearance with more obvious opalescence owing to the NPs enrichment (Figure 3B).

Trehalose could stabilize biological macromlecules (e.g., proteins) in the folded state under conditions that would normally promote their denaturation [31]. Fonte et al. [32] demonstrated that trehalose showed superior insulin stability in insulin-loaded PLGA NPs. Trehalose was selected as a cryoprotectant with albumin-loaded NPs, and the main mechanism was explored regarding cryoprotection [33]. As shown in Figure 3C, comparing the different concentrations of trehalose (*w/v*), it emerged that at a concentration of 5, lyophilized NPs appeared with a greater increase in particle size than at 10 and 15. In contrast, the increase in particle size was less than 50 nm and the PDI was less than 0.25 when the concentration of trehalose was 10% and 15%, suggesting that a complete re-dispersion was obtained. Furthermore, the effect of trehalose on the formulation at a concentration of 15% was studied by analyzing the TEM images (Figure 3D), indicating that the particle size of BSA-loaded NPs with trehalose was similar to the NPs before lyophilization (Figure 2B). Meanwhile, NP aggregation resulted in higher particle size and PDI in DLS.

Therefore, it can be concluded that for BSA-loaded NPs, the use of trehalose with the appropriate concentration as lyoprotectant is very effective in controlling the particle size of NPs without affecting the physicochemical characteristics after re-dispersion of the freeze-dried samples.

### 2.4. Microstructure of NPs

#### 2.4.1. Fluorescence Quenching of BSA

Tryptophan (Trp) fluorescence quenching assay is considered to be a useful method to provide information on the binding properties of drugs to proteins, such as the quenching mechanisms, binding mechanisms, binding constants, binding patterns, and binding sites [34]. As shown in Figure 4A, the fluorescence quenching spectra produced maximum emission at 338 nm, indicating that the binding sites were close to Trp residues. Meanwhile, with the addition of CS, a noticeable decrease occurred in the fluorescence intensity of BSA, suggesting a direct interaction between CS and BSA. Similar fluorescence quenching was reported upon the interaction of various drugs with BSA [35,36]. The binding kinetics between CS and BSA were further investigated using the Stern–Volmer equation (Equation (1)):(1)$$\frac{F_{0}}{F}=1+K_{SV}\left[Q\right]=1+k_{q}\tau_{0}\left[Q\right]$$
where *F*_0_ and *F* are the fluorescence intensities of BSA aqueous solution in the absence and presence of the quencher, respectively. *Kq* is quenching rate constant. *τ*_0_ is the lifetime of the fluorophore in the absence of quencher and its value is 5.9 ns for BSA aqueous solution [35]. [Q] is the quencher concentration and *Ksv* is the Stern–Volmer quenching constant.

The values of *Kq* and *Ksv* obtained from the slope of Stern–Volmer plots (Figure 4B) are reported in Table 4.

In general, the fluorescence quenching mechanisms are classified as dynamic quenching (collision encounter), static quenching (non-fluorescent complex formation), or combination mechanism [37]. Furthermore, dynamic and static quenching can be distinguished by their different factors in terms of temperature, viscosity, and excited-state lifetime. For dynamic quenching, higher temperature results in faster diffusion collisions and larger quenching constant [38]. In this study, the *Ksv* value was positively correlated with the rising temperature, which may indicate that dynamic quenching is involved in the quenching process. Besides, the *Kq* value was greater than the maximum diffusion collision quenching rate constant (2 × 10^10^ M^−1^ s^−1^), indicating the presence of static quenching during the binding process. Therefore, we can perorate that the fluorescence quenching mechanism of BSA with CS is triggered by a combination mechanism.

#### 2.4.2. Interaction Parameter and the Combination Types

The non-covalent interactions between polysaccharides and proteins in aqueous solutions can be driven by hydrogen bonding, hydrophobic and electrostatic interactions [39]. In order to elucidate the nature of the binding force between CS and BSA, the fluorescence data was also used to determine the thermodynamic parameters of the complex by the following equations (Equation (2)):(2)$$log\left[\frac{\left(F_{0}-F\right)}{F}\right]=logK_{b}+nlog\left[Q\right]$$
where *Kb* is the binding constant, and *n* is the number of binding sites.

The value of Δ*H*_0_ can be regarded as a constant when there is no obvious change in temperature. Then the values of Δ*H*, Δ*S*, and Δ*G* can be evaluated by the van’t Hoff and thermodynamic equations (Equations (3) and (4)):(3)$$lnK_{b}=-\frac{\Delta H}{RT}+\frac{\Delta S}{R}$$
(4)ΔG=ΔH−TΔS
where *R* is the gas constant. The experimental temperatures used were 298, 303 and 308 K.

The results are tabulated in Table 5. The sign of ∆*G* determines the feasibility of the binding reaction between polymer and protein. If Δ*H* > 0 and Δ*S* > 0, the main force is a hydrophobic force. If Δ*H* < 0 and Δ*S* > 0, the electrostatic force dominates the interaction. Finally, if Δ*H* < 0 and Δ*S* < 0, both van der Waals forces and hydrogen bonds are established [40]. The negative sign of *∆G* indicates the spontaneous binding of CS to BSA. Table 5 shows ∆*H* > 0 (407.40 kJ mol^−1^) and ∆*S* > 0 (1.43 kJ mol^−1^ K^−1^), implying that hydrophobic force is the main force between CS and BSA. Furthermore, it is generally agreed that if *Kb* for the interaction of BSA with various substances is in the range of 115 × 10^4^ M^−1^, the binding affinity is moderate [41]. Interestingly, at 308K, the binding constant of CS and BSA was 5.988 × 10^5^ M^−1^, and *Kb* increased with higher temperature, suggesting that the binding affinity of CS and BSA was high under the preparation conditions.

#### 2.4.3. Fourier Transform Infrared Spectroscopy (FT-IR)

FT-IR was used to determine variations in chemical functional groups in the samples (Figure 4A). Generally, according to the previously published literature [42,43,44], CS (Figure 4C(a)) showed specific peaks at 3354.9 (–OH and –NH_2_ stretching), 2873.2 (–CH stretching), 1645.0 (amide I, C O stretching), 1588.4 (amide II, N–H bending), 1419.8 (–CH_2_ bending), 1375.3 (–CH_3_ symmetric deformation), 1026.1 (C–O–C stretching), and 895.1 cm^−1^ (glucose ring) [45]. Compared with the spectrum of CS, the peak of blank NPs (Figure 4C(b)) shifted from 3354.9 to 3249.2 cm^−1^ and became wider and flatter, indicating that hydrogen bonding was enhanced. The peaks of amide I and amide II in blank NPs shifted to 1633.3 and 1535.8 cm^−1^, respectively, due to the electrostatic interactions between phosphoric groups of TPP and amino groups of CS in NPs. These observations were consistent with the results reported previously [46,47]. BSA (Figure 4C(c)) showed specific peaks at 3283.9 (–NH_2_ stretching), 2930.1 (–CH stretching), 1638.6 (amide I), 1532.6 (amide II), and 1391.5 cm^−1^ (amide III, C–N stretching). In comparison with blank NPs, the BSA-loaded NPs (Figure 4C(d)) showed a significant increase in the intensity of the amino acid characteristic peak at 1650–1500 cm^−1^, and the peak shape was similar to BSA, reflecting the successful encapsulation of BSA in NPs.

#### 2.4.4. CD Spectra Studies

CD spectroscopy has been widely used to determine the secondary and tertiary structure of proteins in solution, which is sufficiently precise for biopharmaceutical protein characteristics [48]. Therefore, CD analysis is sensitive enough to detect changes in protein content [49]. The experimental CD spectra of the BSA-loaded NPs suspension and supernatant are presented in Figure 4D. BSA is predominantly α-helical, as expected, and the CD spectrum exhibits a strong negative band (ellipticity) at 240–200 nm [50]. Previous studies [51,52] have reported that the manufacturing process of NPs could potentially induce an alteration of the molecular conformation of proteins in such NPs. However, in this study, the obviously raised ellipticity of BSA-loaded NPs suspension, which is an indicator of α- helical structure of BSA, demonstrated that this ionic gelation process is adequately friendly to the BSA and did not change its molecular structure. In this study, the increase of the alpha-helix of BSA in CS NPs may come from the strong hydrophobic force between CS and BSA [53,54,55]. However, more thorough research is necessary to further explain this interesting fact directly and scientifically.

### 2.5. In Vitro Release of NPs

In order to investigate the feasibility of CS-NPs as drug carriers for therapeutic proteins, an in vitro release study of BSA-loaded NPs was performed. Previous studies have reported that drug molecule diffusion plays a predominant role in release profile when the particle size of an encapsulated drug molecule is much smaller than the formed NPs [46]. Proteins are macromolecules, some of which react with CS [56,57]; therefore, the release process is particularly complex.

Analysis of BSA release showed a very rapid initial burst (0–12 h), followed by a slow release in the samples (Figure 5A). Gan and Wang’s research shows that after the burst release of protein, considerable particles swelled with loss of density, increase in particle size, and loss of physical integrity. So Gan and Wang concluded that the burst release of BSA is more likely a consequence of rapid surface desorption of large amounts of protein molecules from a huge specific surface area provided by large numbers of NPs [19]. Consistent with FT-IR spectra that the BSA-loaded NPs decreased intensity in the main characteristic bands of carbonyl (C=O–NHR) and amine groups (–NH_2_) at 1636 cm^−1^ and 1540 cm^−1^ before and after release at the first 12 h [44], the degree of deacetylation of chitosan used in this study is high (75–85%), leading to a denser formed nanostructure due to the more available cations and anions. It was then shown that BSA is slowly released after 12 h because protein macromolecules are difficult to release from the pores of the NPs produced by polymer erosion. The high binding affinity of CS to BSA (in Section 2.4.2) is also a factor responsible for the slow release of BSA. This resulted in the small accumulative release of BSA (44.3% at 144 h) as well.

The integrity of the encapsulated proteins was evaluated by sodium dodecyl sulfate-polyacrylamide gel electrophoresis (SDS-PAGE) analysis. SDS-PAGE analysis followed by Coomassie blue staining revealed identical bands for the native and entrapped proteins (Figure 5B). This indicates that the preparation and freeze-drying conditions did not cause any irreversible aggregation or cleavage of the proteins [58].

### 2.6. Cytotoxicity Analysis

To determine if the lyophilized BSA-loaded NPs were innocuous, the cell viability of L929 cells was measured using the MTT assay. First, L929 cells were added into the 96-well plate at a certain density (dulbecco’s modified eagle’s medium + 10% (*v/v*) fetal bovine serum), and then the lyophilized BSA-loaded NPs solution was added to the 96-well plate at different concentrations. After 4 days of cultivation in the CO_2_ incubator, the cell survival rate was calculated through the absorbance in the microplate reader. The results are presented in Table 6, good cell viability (>80%) was observed for L929 cells containing re-dissolved BSA-loaded NPs with varying concentrations from 10 to 100 μg/mL. These results suggest that CS-NPs with trehalose have excellent biocompatibility and these proved the CS-NPs to be a safe vehicle for the delivery of proteins.

## 3. Materials and Methods

### 3.1. Materials

Low molecular weight chitosan (LMWC) and trehalose dihydrate were purchased from Sigma (St. Louis, MO, USA). Tripolyphosphate pentasodium (TPP), bovine serum albumin from (BSA) and fluorescein 5(6)-isothiocyanate (FITC) were purchased from Aladdin Chemicals (Shanghai, China). Sucrose, mannitol, and d-glucose were purchased from China Pharmaceutical of Shanghai Chemical Reagent Co. (Shanghai, China). Dialysis bags (MWCO = 7000 were obtained from Gene Star Co. (Shanghai, China). All other chemicals and reagents were of the highest purity grade commercially available.

### 3.2. Preparation of the NPs

The blank NPs were prepared using the procedure described by Q. Gan [19] with slight modifications. CS stock solution (3 mg/mL) was prepared by dissolving CS in 0.05% (*v/v*) acetic acid solution and leaving it under stirring for 24 h to produce homogeneous solutions. The pH was adjusted to 5.5 with a 1 M sodium hydroxide solution. CS stock solutions were filtered through a 0.45 µm membrane and diluted in deionized water to the final desired concentrations. TPP solution (0.7 mg/mL) was added to the CS solution dropwise at different CS: TPP ratios (*w/w*) and the dispersions were allowed to stabilize for 60 min under constant vigorous magnetic stirring at room temperature. BSA-loaded NPs were prepared in a similar manner except that different volumes of the BSA solution (1.0 mg/mL) in deionized water were added directly to the CS solution before adding the TPP solution.

### 3.3. NPs Recovery

NPs were centrifuged (15,000 RFC, 4 °C, 60 min) and the obtained sediment was resuspended in deionized water, using lyophilized trehalose, sucrose, mannitol, and glucose as cryoprotectants. Dried NPs were re-dispersed and characterized in terms of particle size, PDI, and morphology [59].

### 3.4. Physicochemical Characteristics of NPs

#### 3.4.1. Particle Size and Morphology

The particle size and *Zeta* potential of the NPs were determined at 25 °C by dynamic light scattering (DLS) (Zetasizer Nano ZS90, Malvern, UK). Particle size values were reported as the mean hydrodynamic diameter (MHD), standard deviation (SD), and polydispersity index (PDI). Each batch was analyzed in triplicate. The morphology of NPs was examined by transmission electron microscope (TEM) (JEM-1010, JEOL, Tokyo, Japan). Briefly, fresh made NPs samples were first diluted with pure water. A drop of diluted sample was dripped onto a copper grid and allowed to stain negatively by 1% uranyl acetate solution. Then, the sample was air-dried at room temperature before viewing on the TEM.

#### 3.4.2. The Encapsulation Efficiency and Loading Capacity of NPs

EE of BSA was calculated from the amount of BSA remaining in the supernatant collected upon centrifugation of the NPs. The concentration of BSA was determined by the standard Micro-BCA protein assay. The supernatant of unloaded NPs was used as a blank, in order to subtract the interference from the components of loaded samples with the BCA. LC was calculated from the weight of sediments collected after centrifugation. EE and LC were calculated using the following equation:(5)$$EE\left(\%\right)=\frac{C_{total}-C_{Super}}{C_{total}}\times 100\%$$
(6)$$LC\left(\%\right)=\frac{\left(C_{total}-C_{super}\right)\times V}{W}\times 100\%$$
where *C_total_* and *C_super_* are the concentration of BSA in NPs and in the supernatant. *V* is the volume of NPs. *W* is the weight of NPs.

### 3.5. NPs Release and Stability Studies

#### 3.5.1. In Vitro BSA Release Studies

FITC-BSA was prepared using the method provided by Sigma-Aldrich with slight modifications. In brief, BSA was dissolved in 0.1 M sodium carbonate buffer (pH 9), then FITC (1 mg/mL) in anhydrous DMSO was added to the BSA solution (1 mg/mL) dropwise under gentle stirring. Finally, the reaction was incubated in the dark for 12 h at 4 °C. The conjugate was placed into a dialysis bag for 3 days to remove DMSO and unbound FITC, and then lyophilized to obtain FITC-BSA powder.

For BSA release studies, lyophilized FITC-BSA-loaded NPs were transferred to a beaker containing 200 mL of PBS (pH 7.4). The beaker was placed in an incubator at 37 °C ± 0.5 °C with shaking at 100 rpm. At designated time intervals, 1 mL was withdrawn from the beaker and replaced with an equivalent volume of fresh PBS. Samples were centrifuged and the amount of BSA released was determined by measuring the fluorescent density of FITC-BSA (excitation wavelength: 490 nm, emission wavelength: 540 nm) using a hybrid multi-mode microplate reader (Synergy™ H1, BioTek, Winooski, VT, USA).

#### 3.5.2. Sodium Dodecyl Sulfate-Polyacrylamide Gel Electrophoresis

The structural integrity of BSA-loaded in NPs was evaluated by sodium dodecyl sulfate-polyacrylamide gel electrophoresis (SDS-PAGE) analysis [58,59,60]. Briefly, lyophilized BSA-loaded NPs were incubated for 24 h in PBS (pH 7.4) at 37 °C under mild shaking (100 rpm). The BSA released from NPs was separated by centrifugation. The supernatant was collected and suspended in SDS-loading buffer. The samples were run on 8% SDS-PAGE gel at 180 V until the dye band reached the gel bottom. Proteins were visualized by Coomassie blue staining.

### 3.6. Multi-Spectroscopic Investigation of NPs

#### 3.6.1. Fluorescence Spectroscopy Measurements

Fluorescence spectroscopy (F96s, Shanghai Lengguang technology company, Shanghai, China) was used to determine the changes in tryptophan (Trp) fluorescence at 298, 303 and 308 K, which was induced by the binding of BSA to CS [61]. Trp fluorescence quenching experiments were carried out as follows: CS and BSA were dissolved in Tris-HCl buffer (pH 5.5). Then, CS solution was added to the BSA solution, and the final concentrations varied from 0 to 0.5 µM. The fluorescence emission spectra were recorded in the wavelength range of 300–500 nm upon excitation at 285 nm. The data, which collected after inner filter correction, was used to calculate the quenching parameters. The overall concentration of BSA was maintained at 1 µM, and all the experimental data for spectroscopic measurements were the average of three sets of repetitions.

#### 3.6.2. FT-IR Measurements

Two milligrams of dried samples were mixed evenly in 80–100 mg KBr powder and characterized using Fourier Transform Infrared-attenuated total reflectance spectroscopy. CS, BSA, blank NPs, and BSA-loaded NPs were analyzed using FT-IR spectrophotometer (Nicolet 6700, Thermo Fisher, Waltham, MA, USA). Signal averages were obtained from 64 scans in the frequency range of 4000–400 cm^−1^ at a resolution of 4 cm^−1^.

#### 3.6.3. CD Spectra Measurements

Re-dissolved BSA-loaded NPs suspension and the supernatant which collected upon centrifugation were analyzed using circular dichroism Spectrophotometer (JASCO J-815, Japan Spectroscopic Company, Tokyo, Japan) with 0.1 cm quartz cell at room temperature. The Xe lamp was selected after 10 min of high-purity nitrogen with a flow rate of 100 L/h. The CD spectra were measured at a scanning rate of 100 nm/min in the wavelength range of 200–300 nm with an interval of 1 nm. With the blank NPs suspension as control group, each spectrum was scanned three times and the average of the three scans was utilized.

### 3.7. Cytotoxicity Assay

The cytotoxicity of lyophilized BSA-loaded NPs was evaluated with the MTT assay using L929 mouse embryonic fibroblasts. Cells were cultured in Dulbecco’s modified Eagle’s medium supplemented with 10% (*v/v*) fetal bovine serum in 96-well plates. The cells were exposed to various doses of sterilized re-dissolved BSA-loaded NPs or medium only (negative control) at 37 °C for 4 days. A 20 µl volume of MTT labeling reagent was added, and cells were cultured for 4 h at 37 °C. The absorbance was measured at 570 nm, and cell viability (%) was calculated as follows:(7)$$Cell\;viability\left(\%\right)=\frac{{\left[A\right]}_{test}}{{\left[A\right]}_{control}}\times 100\%$$
where [*A*]test and [*A*]control are the absorbances of the test and negative control solutions, respectively.

### 3.8. Statistical Analysis

Data were analyzed by one-way ANOVA among multiple groups or a Student’s t-test between two groups with Prism Graph Pad software (Version No.7.0, Graph-Pad Software, La Jolla, CA, USA). A value of *p* < 0.05 was considered statistically significant. Results are reported as the mean ± SD for triplicate measurements.

## 4. Conclusions

In summary, CS and TPP spontaneously formed the basic skeleton of the NPs through electrostatic interaction. Then, BSA was incorporated into the basic skeleton by non-covalent interaction-hydrophobic forces, and the structure and function of BSA was not changed. At the same time, some of the BSA adsorbed on the surface of NPs (part of which were inlaid on the NPs) were exposed to the solution to give the burst release. This work demonstrated that protein-loaded CS-NPs is promising to effectively improve protein stability in vitro and to enhance the effects of proteins in vivo owing to the delayed-release and controlled-release of NPs. Moreover, utilizing the DHPM could not only effectively reduce the reaction time and efficiently increase the productivity, but also dramatically reduce the risk of contamination during the manufacturing processes Those enable the DHPM to be a promising process to produce versatile nanoparticles for clinical applications.

## Figures and Tables

**Figure 1 marinedrugs-18-00315-f001:**
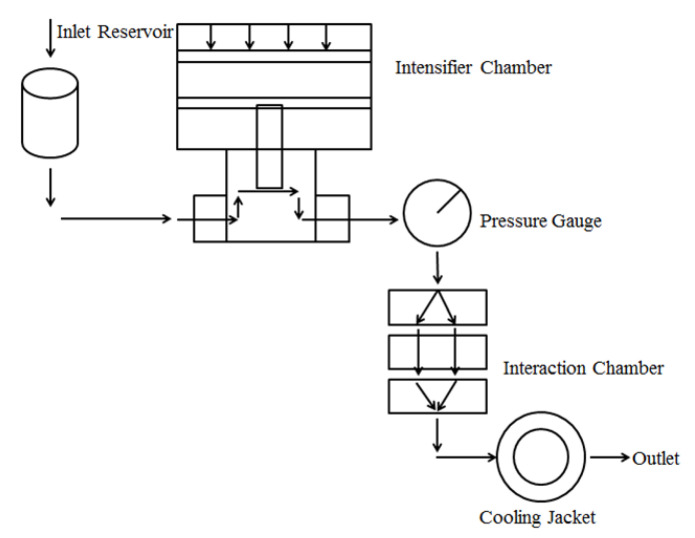
The principle of DHPM. The NPs solution enters the intensifier chamber from the inlet and then enters the cooling chamber through the interaction chamber to achieve dispersion, homogenization, emulsification, etc. The pressure of the instrument chamber was adjusted by the pressure gauge.

**Figure 2 marinedrugs-18-00315-f002:**
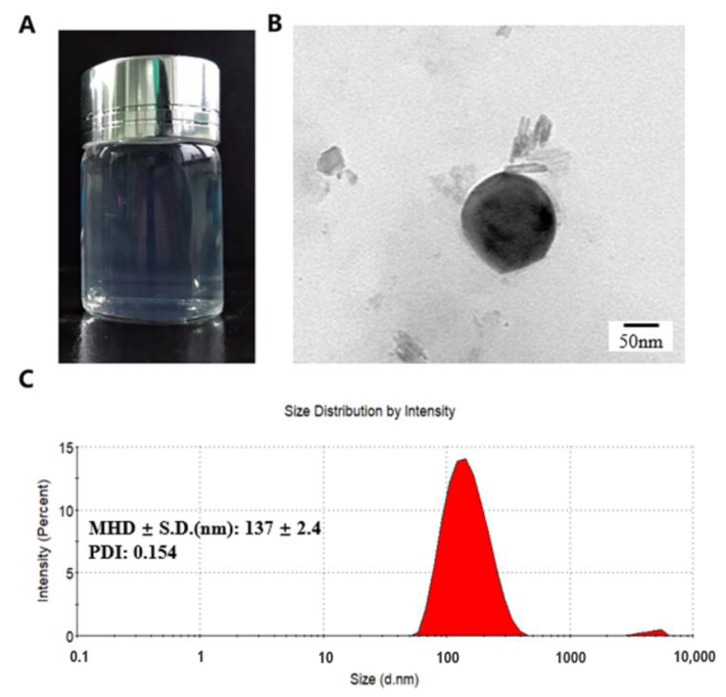
(**A**) Optic image of BSA-loaded NPs; (**B**) TEM image of BSA-loaded NPs; (**C**) Corresponding DLS image (CS concentration = 1 mg/mL, CS: TPP ratios (*w/w*) = 3:1). Data were expressed as mean ± standard error.

**Figure 3 marinedrugs-18-00315-f003:**
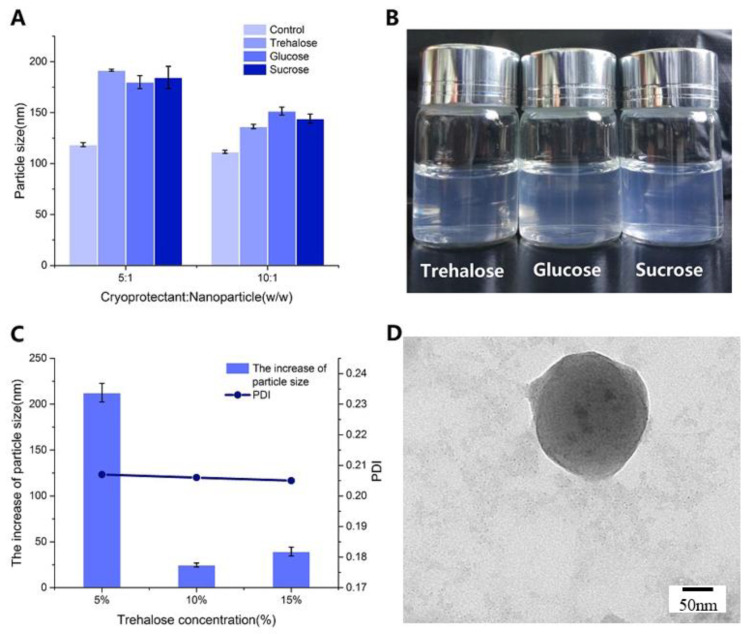
(**A**) The Effect of cryoprotectants on particle size. The particle size of untreated NPs (control) and re-dissolved BSA-loaded NPs lyophilized with trehalose, glucose or sucrose as cryoprotectants (cryoprotectants concentration = 10% (*w/v*)); (**B**) Optic image of re-dissolved BSA-loaded NPs (cryoprotectants concentration = 10% (*w/v*), cryoprotectants: nanoparticles (*w/w*) = 10:1); (**C**) The increase of particle size of re-dissolved BSA-loaded NPs (cryoprotectants concentration = 5%, 10% and 15% (*w/v*), cryoprotectants: nanoparticles (*w/w*) = 10:1); (**D**) TEM image of re-dissolved BSA-loaded NPs (trehalose concentration (*w/v*) = 15%, trehalose: nanoparticles (*w/w*) = 10:1). (CS concentration = 1 mg/mL, CS: TPP ratios (*w/w*) = 3:1). Values in (a), (c) were expressed as mean ± SD.

**Figure 4 marinedrugs-18-00315-f004:**
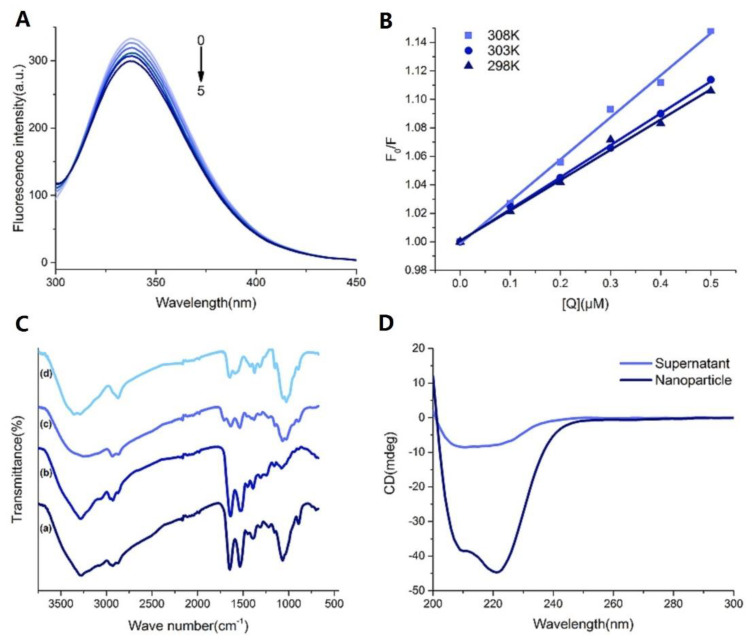
(**A**) The fluorescence emission spectra of BSA (1.0 µM) in the presence of different amounts of chitosan (T = 298 K, pH = 5.5). C (chitosan) = 0, 0.1, 0.2, 0.3, 0.4, 0.5 µM (0–5); (**B**). The Stern–Volmer plot for chitosan–BSA interactions at different temperatures; (**C**). Fourier transform infrared (FT-IR) spectra of (a) chitosan, (b) blank NPs, (c) BSA and (d) BSA-loaded NPs. (CS concentration = 1 mg/mL, CS: TPP ratios (*w/w*) = 3:1); (**D**). Circular dichroism (CD) spectra of re-dissolved BSA-loaded NPs suspension and supernatant corrected by blank NPs suspension or supernatant, respectively. (CS concentration = 1 mg/mL, CS: TPP ratios (*w/w*) = 3:1, trehalose concentration (*w/v*) = 15%, trehalose: nanoparticles (*w/w*) = 10:1).

**Figure 5 marinedrugs-18-00315-f005:**
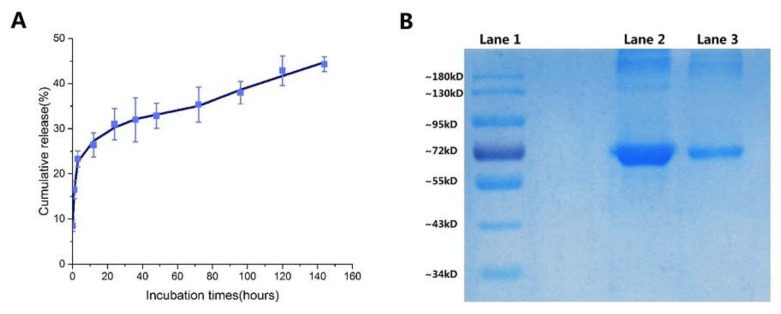
(**a**) BSA release profiles from the lyophilized BSA-loaded NPs; (**b**) SDS-PAGE analysis: (Lane 1) marker proteins, (Lane 2) native BSA; (Lane 3) BSA release from lyophilized BSA-loaded NPs. (CS concentration = 1 mg/mL, CS: TPP ratios (*w/w*) = 3:1, trehalose concentration (*w/v*) = 15%, trehalose: nanoparticles (*w/w*) = 10:1). Data were expressed as mean ± standard error.

**Table 1 marinedrugs-18-00315-t001:** The Physicochemical Characteristics of CS-NPs prepared by using different CS/TPP mass ratios.

CS: TPP (*w/w*)	MHD ± SD (nm)	PDI	ζ-Pot ± SD (mv)	EE ± SD (%)	LC (%)
3:1	92 ± 2	0.14	16 ± 0.4	53 ± 5	67 ± 1
4:1	99 ± 2	0.21	20 ± 0.5	34 ± 3	56 ± 2
5:1	118 ± 2	0.27	25 ± 0.2	28 ± 1	47 ± 1
6:1	120 ± 1	0.26	26 ± 0.2	20 ± 2	/
blank NPs (3:1)	112 ± 4	0.23	19 ± 0.5	/	/

CS concentration = 1 mg/mL; Results were reported as mean ± SD, *n* = 3.

**Table 2 marinedrugs-18-00315-t002:** The Physicochemical Characteristics of CS-NPs prepared by using different pressures.

Pressure (psi)	MHD ± SD (nm)	PDI	ζ-pot ± SD (mv)	EE ± SD (%)	LC (%)
10	98 ± 2	0.14	18 ± 0.3	82 ± 1	9 ± 1
20	89 ± 2	0.12	19 ± 0.2	82 ± 1	18 ± 17
30	89 ± 2	0.10	21 ± 0.5	83 ± 1	33 ± 2
Coarse NPs	92 ± 2	0.14	16 ± 0.4	53 ± 5	67 ± 2 *

CS: TPP (*w/w*) = 3:1; Results were reported as mean ± SD, *n* = 3; *: The LC of coarse NPs reached 66.9%, which is beyond the normal range. This is because the electronic balance error is caused by the low single preparation of nanoparticles. The LC of post-treated NPs with DHPM reached 33.4%, which is within the normal range.

**Table 3 marinedrugs-18-00315-t003:** The Physicochemical Characteristics of CS-NPs prepared by using different cycle times.

Cycle Times(min)	MHD ± SD (nm)	PDI	ζ-pot ± SD (mv)	EE ± SD (%)	LC (%)
2	89 ± 2	0.10	21 ± 0.5	93 ± 1	33 ± 1
4	88 ± 2	0.11	22 ± 0.1	85 ± 2	36 ± 1
6	90 ± 2	0.10	21 ± 0.1	84 ± 1	37 ± 2
8	89 ± 0.2	0.12	20 ± 0.2	87 ± 1	40 ± 1
10	92 ± 2	0.13	19 ± 0.2	85 ± 13	38 ± 1

CS: TPP (*w/w*) = 3:1; Pressure = 30 psi; Results were reported as mean ± SD, *n* = 3.

**Table 4 marinedrugs-18-00315-t004:** The Stern–Volmer quenching constant and bimolecular quenching rate constant for CS-BSA complex at different temperatures.

*T*(K)	Slope	Intercept	r^2 a^	*Ksv* × 10^5^ (M^−1^)	*Kq* × 10^13^ (M^−1^ s^−1^)
298	0.21	1.00	0.9922	2.13	3.61
303	0.22	1.00	0.9994	2.24	3.80
308	0.30	1.00	0.9955	2.95	5.00

r^2 a^ is the correlation coefficient.

**Table 5 marinedrugs-18-00315-t005:** The Stern–Volmer double logarithmic regression parameters of CS–BSA complex at different temperatures.

*T*(K)	Log*Kb*	*n*	r^2 a^	*Kb*(M^−1^)	Δ*G* (kJ.mol^−1^)	Δ*H* (kJ.mol^−1^)	Δ*S* (kJ mol^−1^ K^−1^)
298	3.4624	0.9084	0.9984	2.900 × 10^3^	−18.74	407.40	1.43
303	4.9684	0.9398	0.9954	9.298 × 10^4^	−25.89		
308	5.7773	1.0673	0.9969	5.988 × 10^5^	−30.44		

r^2 a^ is the correlation coefficient.

**Table 6 marinedrugs-18-00315-t006:** Cytotoxicity evaluation of lyophilized BSA-loaded NPs for different concentrations.

Sample	BSA Concentration (µg/mL)	Cell Viability (%)
lyophilized BSA-loaded NPs	100	82 ± 2
	50	90 ± 2
	10	93 ± 3

Data were expressed as a means ± standard error.

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
