# Peer review of "Characteristics, Cryoprotection Evaluation and In Vitro Release of BSA-Loaded Chitosan Nanoparticles"

_marinedrugs, 2020, doi:10.3390/md18060315_

Round 1

Reviewer 1 Report

The article “Characteristics, microstructure, cryoprotection evaluation, and in-vitro release of BSA-loaded chitosan nanoparticles” is an interesting work adequate for publishing in Marine Drugs (MDPI) but some modifications must be carried out to be accepted.

1.- In title it is said “microstructure” but there is no any microstructure analysis, there only appears some TEM photographs showing the spherical shape of the nanoparticles. The microstructure involves SEM, pore size distributions, surface area, topology, etc. analysis. This word must be removed.

2.- There is a confusion in Figures because the second Figure 2 must be Figure 3.

3.- Most of all data on Tables contain a lot of decimals not corresponding with correct errors of instruments or data analysis (calculus, slopes straight lines, etc.). This must be checked in all Tables.

4.- How are measured the FTIR spectra? By the ATR technique? By dilution in KBr? This is important to understand the 4000-3000 cm-1 spectral region.

5.- In Section 2.4.3. FTIR. With a 4.0 cm-1 resolution it is not adequate to use decimals in peak positions.

6.- Figure 3C (although is text is named Figure 2c), how is calculated the increase of particle size? Nor the instrument and technique, but the sample which is used as reference. This must be described because in Figure 3A all values are between 110-190 nm, while Figure 3c are close to 25 nm.

7.- Figure 3d (although is named as Figure 2d) what does mean “data were expressed as mean+/- standard error? This is written in the text of the figure 3d.

8.- Most cases the particle size of nanoparticles are expressed as “size” but it must be correctly written as “particle size” in all cases.

9.- A lot of times numbers are expressed by using one decimal that is not significant, for example 4000.0 cm-1 to 400.0 cm-1, or 400.0 rpm, 540.0 nm, 166,000.0, etc., while other times no decimal is used. All of these must be checked.

10.- In page 3- Line 101: Molecular weight will be referred as MW no Mn (this is Manganese)

11.- In page 9. Kb is two times described (in eq.2 and in eq.3). Only one is necessary. And in the next paragraph is also described and not abbreviated. This must be modified.

12.- In page 9- following eq.2, in next line: n must be in italic.

13.- Encapsulation efficiency is abbreviated as EE but there are several times that is completely written in the text.

14.- In page 6-Line 192. Define what “amorphous glass” is.

15.- In page 7-Line 206: Pedro Fonte et al., must be written as Fonte et al.

16.- In all the text there are (probably) some grammar mistakes that must be checked. For example, page 4, lines 127 to 132 are difficult to understand. The last paragraph of Page 10 and beginning in Page 11 : probably “physical integrity. So Can and …” the dot must be comma. In the text of Figure 1 it is written “to achieves”. In page 2/ Line 54: “Most CS present in the cuticles…” probably must be “Most CS is present…”. In page 3 – Line 100: impurities, So we … must be “so” without capital letter. Etc.

Reviewer 2 Report

The authors are describing BSA loaded chitosan nanoparticles and its properties

Following points need to be addressed:

  1. What is the novelty and innovation of the study? Authors need to highlight about it in the abstract and the introduction section.
  2. How does the study advance the knowledge in the field and is different from the ones already reported?
  3. I do not think the studies are enough for acceptance into marine drugs journal and authors need to enhance the potential of their study through additional experiments
  4. Similar studies have already been reported and in much advanced manner and the study lacks novelty, innovation and scientific advancement

Reviewer 3 Report

In this paper, the authors have prepared chitosan nanoparticles for the delivery of therapeutic proteins by an ionic gelation method, and BSA was used as a model protein. The obtained BSA-loaded chitosan nanoparticles have been characterized involving some techniques and BSA release studies. To improve the long-term stability of NPs, the authors have investigated some cryoprotectant as trehalose, glucose, sucrose, and mannitol were selected Furthermore, they have used trehalose to obtain re-dispersible lyophilized NPs which significantly reduce the dosage of cryoprotectants. Although the experimental part seems to me correctly performed, the paper needs, in my opinion, a major revision before to be accepted for publication.

1.- Lines 120-122: the data of the dependence of BSA EE on the concentration of the CS in the NPs should be shown.

2.-  Table 1 (lines 123-126): i) Given the value of PDI, it is strange that the SD values for the sizes are so small. Could the authors explain how they got the SD values?  ii) The authors should express the data (size, Z-potential, etc.) in the table with the correct significant figures (for example, 53±5 instead of 52.66±4.98). iii) Why have the authors not determined the LC for the NPs CS/TPP 6:1? If it has not been determined, the authors should comment on the reason for this. iv) The value of PDI for the blank NPS (4.233) must be a mistake. As is well know PDI is in the range [0,1].

3.-  Table 1 (lines 160-124): i) Given the value of PDI, it is strange that the SD values for the sizes are so small. Could the authors explain how they got the SD valees?  ii) The authors should express the data (size, Z-potential, etc.) in the table with the correct significant figures (for example, 83±1 instead of 82.858±1.330).

4.- Figures 2B and 3B: TEM images are of very poor quality. The uthors should present other better images.

5.- Figure 2 C: Why do authors use the number size distribution instead of intensity? Is it because they don't see any aggregates? I would like to see the representation in intensity? I know that with pdi<0.2 the same value of size in the two systems would be observed, is that the case?

6.- Line 220: Figure 2 must be Figure 3.

7.- Section 2.4.1. Fluorescence quenching of BSA (the lines in this section are not numbered in the pdf document): i) The authors sometimes represent the quenching rate constant as kq and others as Kq. They must standardize the nomenclature in all the manuscript. ii) Have the authors measured the lifetime of the BSA? If not, they should reference where they have taken the data.

8.- Section 2.4.2. Interaction Parameter and the Combination Types (the lines in this section are not numbered in the pdf document): i) The authors sometimes represent the energetic terms as ΔH0, ΔH0, ΔH, ΔG0, ΔG0, ΔG, ΔS0, ΔS0 and ΔS. They must standardize the nomenclature in all the manuscript: in my opinion it is better without 0 because the Kb is not a true thermodynamic constant. ii) There is a mistake in Table 5: the units of ΔS are kJ mol-1 K-1.

9.- Section 2.4.3. Fourier transform infrared spectroscopy (FT-IR) (the lines in this section are not numbered in the pdf document): the authors must give a basis to the assignments of each peaks in these spectra with a several references.

10.- Section 2.4.4. CD spectra studies (the lines in this section are not numbered in the pdf document): Could the authors provide a scientific explanation for the fact that BSA increases the alpha-helix form in the presence of CS in BSA-loaded chitosan nanoparticles?  This phenomenon is also observed when some peptides interact with other systems such as liposomes (see, for example, Biochimica et Biophysica Acta 1828 (2013) 1863–1872).

11.- Section 2.5. In vitro release of NPs (the lines in this section are not numbered in the pdf document): i) The authors say “The sequential time frame of TEM images showed that the BSA-loaded NPs maintained their shape, size and integral structure during the burst release of BSA” Where are these results? Have they performed these experiments? If they have, why don't the results appear? This should be clarified. ii) Where are the results of FT-IR spectra after release at the first 12 h? The authors mention these results in the discussion of the results of the release experiment but they do not appear in the manuscript. iii) The release of the BSA is not complete. Is this because the release has not been followed any longer or is there a reason why the BSA is not fully released from the NPs? This should be clarified.

12.- In all the experimental section: all experiments should be described in more detail, for example, i) the concentrations of the suspensions used should appear in the experimental section in order to describe the experiment properly, regardless of whether they appear in the figure captions or in the text of results and discussion, ii) the CD experiment is not well described, etc. (in general, all the experiments).

Reviewer 4 Report

English revision is required. Some abbreviation explanation is missing (TPP, TRP and others...)

Encapsulation efficiency calculations: what is initial loading concentration of BSA? 

Round 2

Reviewer 2 Report

In my view as indicated before I do not recommend the publication of this manuscript because of lack of novelty, innovation and the already existing study in the field. 

In addition the authors have not made an attempt to address my comments from the previous review

Reviewer 3 Report

Although the changes made to the manuscript have improved it, some of the reviewer's suggestions and questions have not been taken into account. For this reason, the paper needs a major revision before to be accepted for publication.

1.-  Table 1 (lines 131-133): Why have the authors not determined the LC for the NPs CS/TPP 6:1? If it has not been determined, the authors should comment on the reason for this.

2.- Lines 255 and 256: the authosr say “τ0 is the lifetime of the fluorophore in the 255 absence of quencher and its value is 5.9 ns for BSA…” It is missing to indicate in which medium the BSA has this lifetime.

3.- Lines 329-333: Could the authors provide a scientific explanation for the fact that BSA increases the alpha-helix form in the presence of CS in BSA-loaded chitosan nanoparticles?  The authors have not answered this question; they have only described what is observed experimentally.

4.- In all the experimental section Lines 382-471: the experiments should be described in more detail, for example, i) the concentrations of the suspensions used should appear in the experimental section in order to describe the experiment properly, regardless of whether they appear in the figure captions or in the text of results and discussion, ii) the CD experiment is not well described, etc. (in general, all the experiments).

Minor points

1.-  Table 1 (lines 131-133) and 2 (lines 166-167): The authors should express the results of LC in the table with the correct significant figures too.

2.- Line 293: change ΔG0 for ΔG in eq. 4.

3.- The authors should express ΔH and ΔS with the same significant figures in the text (line 299) and in the Table 5 (line 306). The authors must express ΔS in the appropriate units kJ mol-1 K-1 and not J mol-1 K-1 (line 299) or kJ mol1 k-1 (Table 5).

4.- Line 406: change the tittle of this section. It is  unusual to use abbreviations in a title and also the authors must include the encapsulation effiency too.

5.- Supplementary materials (Table S1): The authors should express the data (size, Z-potential, etc.) in the table with the correct significant figures as in the manuscript.

Round 3

Reviewer 3 Report

The changes made to the manuscript have improved it. For this reason, I recommend the article to be accepted for publication in the journal in the present form.